# Clinical and Genetic Characteristics of Parkinson’s Disease Patients with Substantia Nigra Hyperechogenicity

**DOI:** 10.3390/ijms26125492

**Published:** 2025-06-08

**Authors:** Łukasz Milanowski, Piotr Szukało, Małgorzata Kowalska, Alicja Sikorska, Dorota Hoffman-Zacharska, Dariusz Koziorowski

**Affiliations:** 1Department of Neurology, Faculty of Health Science, Medical University of Warsaw, 02-091 Warsaw, Poland; malgorzata.kowalska78@gmail.com (M.K.); dariusz.koziorowski@wum.edu.pl (D.K.); 2Student Scientific Group, Department of Neurology, Faculty of Health Science, Medical University of Warsaw, 02-091 Warsaw, Poland; s080280@student.wum.edu.pl (P.S.); alicja.sikorska@gmail.com (A.S.); 3Department of Medical Genetics, Institute of Mother and Child, 01-211 Warsaw, Poland; dorota.hoffman@imid.med.pl

**Keywords:** Parkinson’s Disease, genetics, hyperechogenicity of substantia nigra, transcranial ultrasonography

## Abstract

Hyperechogenicity of the substantia nigra (SN) is observed using transcranial ultrasonography in patients with Parkinson’s Disease. In this study, we investigated whether monogenic forms of PD are more prevalent in these patients and clinically defined their characteristics. Eighty-eight PD patients were part of the analysis. All patients received clinical diagnoses from experienced movement disorder specialists. Each patient underwent transcranial ultrasonography and genetic testing for mutations in the *SNCA*, *PRKN*, *LRRK2*, *DJ1*, and *PINK1* genes. SN hyperechogenicity was identified in 48 patients. Compared to the non-hyperechogenicity group, these patients did not have monogenic forms of PD more frequently, but they did have REM sleep behavior disorder significantly more often, lived in rural areas, and experienced a later age of disease onset. Our study indicated no association between substantia nigra echogenicity and the presence of mutations in the *SNCA*, *LRRK2*, *DJ1*, *PRKN*, and *PINK1* genes. Hyperechogenicity of the substantia nigra, however, remains a common finding in patients with Parkinson’s Disease, correlating with certain features of the disease.

## 1. Introduction

Parkinson’s Disease (PD) is a progressive neurodegenerative movement disorder characterized by the degeneration of the dopaminergic neurons in the substantia nigra pars compacta and the accumulation of aggregated alpha-synuclein protein in Lewy bodies identified within the surviving neurons. The loss of dopamine signaling in the striatum presents as parkinsonism—a clinical syndrome that includes movement disorders such as tremor, rigidity, bradykinesia, hypokinesia, and akinesia [1]. The diagnosis of the disease is based on clinical symptoms, as no specific test for PD has been developed yet.

The pathogenesis of the most common idiopathic form of PD is the result of the interaction of environmental factors and genetic susceptibility. Most patients develop PD due to the combined effects of several common variants with smaller impacts (detectable via genome-wide association study (GWAS) panels) and environmental influences [2,3]. Among the environmental factors linked to PD occurrence are age [4] and exposure to pesticides and heavy metals [5]. Other factors, such as tobacco and caffeine intake, may provide a protective effect [6]. Approximately 5–10% of patients, however, develop a “monogenic form of Parkinson’s Disease”, which is caused by pathogenic variants in *SNCA*, *LRRK2*, and *VPS35*, genes exhibiting autosomal dominant inheritance, along with *PRKN*, *PINK1*, and *DJ-1*, genes which follow autosomal recessive inheritance [7]. These rare variants show high penetrance, and nearly all carriers develop the disease [2].

Transcranial ultrasonography (TCS) has been utilized to measure the substantia nigra, a crucial brain region impacted in Parkinson’s Disease (Figure 1). Increased echogenicity in the substantia nigra has been linked to the presence of parkinsonian symptoms, indicating dopaminergic neuron loss. During TCS, the probe is placed at temporal bone windows. In about 10% of patients, the bone window is not accessible, preventing the assessment of the SN using ultrasound machines [8]. In an axial section through the midbrain, the butterfly-shaped hypoechogenic midbrain is easily distinguishable from the highly echogenic basal cisterns. According to published studies, SN hyperechogenicity appears to be a valuable marker for diagnosing PD, regardless of the disease stage and severity. However, it seems unsuitable for monitoring disease progression. SN hyperechogenicity has also been identified in individuals carrying mutations of PD genes (*SNCA*, *PRKN*, *PINK1*, *DJ-1*, *LRRK2*), both in PD patients exhibiting clinical symptoms and in asymptomatic individuals [9,10,11]. Moreover, asymptomatic LRRK2 Gly2019Ser mutation carriers with SN hyperechogenicity are more prone to developing PD in the future [12]. The hyperechogenicity of the substantia nigra was also present more frequently in dopa-responsive dystonia patients, despite its different pathophysiology than PD [13].

Recent reports emphasize the value of TCS as a diagnostic and prognostic tool in the case of PD patients [14]. However, the identification of factors leading to the development of PD in presymptomatic carriers remains an unresolved problem. The purpose of this study was to perform an assessment of whether monogenic forms of Parkinson’s Disease occur more frequently among patients with hyperechogenicity of the substantia nigra. Additionally, we analyzed the clinical factors associated with SN hyperechogenicity in PD patients.

## 2. Results

A total of 88 participants were included in the analysis. The transtemporal bone window was available in 79 patients (88.8%). The hyperechogenicity of the substantia nigra was confirmed in 48 patients (60.8%)—23 on the left side, 15 on the right side, and 10 on both sides. The mean age of the patients with hyperechogenicity was 58.2 ± 13.4 vs. 48.3 ± 11.4 without (*p* = 0.04). The mean age of disease onset with hyperechogenicity was 51.5 ± 13.7 vs. 40.9 ± 13.1 without (*p* = 0.04). One compound heterozygote of *PRKN* was revealed in each group: in one patient with hyperechogenicity, *PRKN* biallelic heterozygous mutations—Exon 4 deletion and a nonsense p.Glu79Ter variant—and in one patient without hyperechogenicity, heterozygous biallelic Exon 2 and Exon 4 deletions. The LRRK2 heterozygous missense variant p.Asn1437His was identified in one patient without hyperechogenicity (genotypes according to HGVS recommendation are described in the Section 4).

The presence of pathogenic mutations was 2.1% vs. 6.5% in the hyperechogenicity and non-hyperechogenicity groups, respectively (*p* = 0.55781). Detailed clinical data were available for 29 patients. REM sleep behavior disorder (RBD) was observed more frequently in patients with hyperechogenicity: 60% (N = 9) vs. 7.1% (N = 1), (*p* = 0.00442) (Figure 2).

Patients living in cities were more common in the non-hyperechogenicity group, 85.7% (N = 12), than in the hyperechogenicity group, 42.9% (N = 6), (*p* = 0.04607) (Figure 2) (Table 1). The calculated power of the statistically significant analysis via the chi-square test was at least 99%.

## 3. Discussion

We comprehensively characterized Parkinson’s Disease patients with and without hyperechogenicity in the substantia nigra. Our study did not reveal any associations of the genetic variants with the ultrasonographic findings. The variants were chosen based on previously well-established monogenic inheritance and previous associations with SN hyperechogenicity [15].

Similar negative findings were observed in Gly2019Ser patients. Brügemann et al. compared 34 LRRK2 Gly2019Ser mutation carriers manifesting PD and 24 non-manifesting mutation carriers and did not find statistically significant differences (*p* = 0.439) [16]. A study performed in patients with *ATP13A2*-associated PD did not reveal any hyperechogenicity in these patients [17].

The increased availability of high-throughput testing will allow for the identification of more monogenic forms and cases of PD. The pathophysiology of the different genetic forms of PD may also differ [18]. It was revealed that the typical alpha-synuclein pathology is rare in *LRRK2*, *PINK1*, and *PRKN* patients, which shows that the biological definition of PD has started to become more important [19]. Therefore, the clinical and radiological characterization of genetically defined PD subtypes may vary, and this needs to be recognized.

The hyperechogenicity of the substantia nigra is a well-established auxiliary test in diagnosing Parkinson’s Disease (PD). In our population, hyperechogenicity was observed in 60.8% of patients, which is lower than in previous studies [20,21]. The prevalence of substantia nigra hyperechogenicity among PD patients is notably high, ranging from 68% to 99%, although small spots or lines of hyperechogenicity can also be observed in healthy individuals [8]. SN hyperechogenicity in PD may be detectable in the very early stages of the disease when clinical symptoms are insufficient for a definitive diagnosis. However, the extent of the echo signal is not correlated with disease severity and remains unchanged throughout the disease’s progression. SN hyperechogenicity was noted in patients with a later onset of the disease. Additionally, it was observed more frequently in patients with RBD and those living in rural areas.

Microglia activation and gliosis formation appear to be the primary mechanisms contributing to substantia nigra hyperechogenicity [22], which remains stable throughout PD [23]. Several earlier studies emphasized the significance of iron accumulation [22], though this hypothesis is considered less critical [24]. It may also correlate with disease duration in certain subgroups of PD patients, such as middle-aged males with non-tremor-dominant subtypes [25]. In our patient cohort, we did not find a correlation between echogenicity and prolonged disease duration. These findings align with the results presented in a previous study [26]. However, the onset of the disease occurred later in PD patients exhibiting SN hyperechogenicity.

The prevalence of RBD in PD patients ranges from 14.6% [27] to 77% [28] in general PD populations. In our work, we show that PD patients with SN hyperechogenicity (SN+) have RBD significantly more frequently than PD patients without hyperechogenicity (SN−). In contrast to our results, we have found only one article that shows SN echogenicity in PD patients with RBD not to be significantly different from its presence in PD patients without RBD [29]. However, there are several papers highlighting the value of SN echogenicity as a biomarker. In idiopathic REM sleep behavior disorder patients, SN hyperechogenicity is correlated with an increased risk of developing neurodegenerative conditions such as PD [30], and when combined with ^123^I-FP-CIT SPECT it may be a useful marker in identifying individuals at a higher risk of developing synucleinopathies [31].

According to published studies, the comparison of patients from rural areas and cities has not identified any differences regarding the extent of SN echogenicity. However, a larger proportion of patients with SN hyperechogenicity live in rural areas [32]. The authors, however, did not assess the statistical power of this finding. In our research, we discovered that patients living in rural areas were more prevalent in the hyperechogenicity group (57.1%) than in the non-hyperechogenicity group (14.3%). This may be explained by the presence of pesticides and toxins used in agriculture in the country, as they may facilitate the occurrence of SN hyperechogenicity

Studies indicate that in patients with monogenic forms of PD, the area of SN echogenicity was significantly larger than that of healthy controls, yet smaller than in idiopathic PD [33,34,35]. This may suggest distinct pathomechanisms within these disease entities. In our study, the presence of pathogenic mutations was 2.1% in the hyperechogenicity group compared to 6.5% in the non-hyperechogenicity group. The frequency of mutations was statistically insignificant between the groups. The genetic variants analyzed did not appear to be specific to substantia nigra hyperechogenicity. One interpretation of the incidence of SN hyperechogenicity is that it may be associated with increased iron deposition, meaning that it reflects ultrasound waves more than the surrounding tissues. This could be independent of genetic variants [36,37]. Other studies do not confirm this first interpretation and instead suggest microglia proliferation [8]. Model studies confirm that the hyperechogenicity of substantia nigra is caused by structural changes within the brain tissue rather than by increased iron concentration [24]. It may also be caused by long-lasting dopamine deficiency [13].

There are several limitations to this study. Firstly, due to the retrospective nature of the study, some patients’ clinical data are missing. Secondly, the strength of the group we included is limited regarding the analysis of clinical data, although it is satisfactory considering the number of patients who underwent genetic testing. However, we conducted a power analysis, which was sufficient for the statistically significant values observed in the clinical data. Moreover, we examined patients for only a limited number of genes, which may have resulted in us missing certain mutations in different regions. The genetic testing, which only included Sanger sequencing and MLPA, may have led to the omission of some variants such as intronic or copy number variants (CNVs) in the analyzed genes. Further investigation is needed to draw more precise conclusions.

## 4. Materials and Methods

### 4.1. Clinical and Ultrasonography Analysis

Diagnosis of PD was made based on the UK Brain Bank diagnostic criteria [38] by experienced movement disorder specialists (D.K. and L.M.). Clinical and demographic data were collected through neurological, neuropsychological, speech therapy, and radiological examinations during hospitalization in the Department of Neurology, Faculty of Health Science, Medical University of Warsaw. The study received approval from the Bioethics Committee of the Medical University of Warsaw and was conducted in accordance with the ethical principles outlined in the Declaration of Helsinki. Transcranial ultrasonography was performed using a Philips device with a 1–5 MHz probe by skilled ultrasonographers (D.K. and M.K.). The probe was positioned next to the transtemporal bone window and focused at a depth of 6–8 cm to image the brainstem. The anatomical region of the SN normally appears hypoechogenic. If a hyperechogenic area was detected, it was manually outlined and assessed planimetrically (significant surface area > 0.2 cm^2^).

### 4.2. Genetic Analysis

For all the patients included in the study, full-length coding sequences of the genes *PRKN* (Exons 1–12), *PINK1* (Exons 1–8), *DJ-1* (Exons 1–6), *SNCA* (Exons 1–6), and the diagnostically recommended exons of the *LRRK2* gene (Exons 30, 31, 34, 35, 41, 48) were analyzed through direct sequencing using the Sanger method or as part of whole-exome sequencing (WES) performed in the Deptartment of Medical Genetics Of Institute of Mother and Child. Additionally, exon-level copy number variation (deletion/duplication) analysis was conducted for all patients using the multiplex ligation-dependent probe amplification method (MLPA) (MRC Holland Salsa P051, P052, Amsterdam, the Netherlands). The analyzed genes and identified variants were those most frequently observed in the European PD patients, especially in the Polish population [39].

The variants identified in the *PRKN* and *LRRK2* genes and patients’ genotypes were described according to HGVS v.21.0.4 recommendations (https://hgvs-nomenclature.org/stable/ accessed on 30 April 2025) and the canonical reference sequences GRCh38 NM_004562.3 MANE Select and NM_198578.4 MANE Select, respectively (https://www.ncbi.nlm.nih.gov/genbank/ accessed on 30 April 2025).

The identified variants in *PRKN* and *LRRK2* have been previously documented in the Human Gene Mutation Database Professional (HGMD Professional 2024.2, https://my.qiagendigitalinsights.com/bbp/view/hgmd/pro accessed on 30 April 2025) and/or ClinVar database (www.ncbi.nlm.nih.gov/clinvar accessed on 30 April 2025).

Variants and patient genotypes according to the HGVS recommendations [https://hgvs-nomenclature.org/stable/ accessed on 30 April 2025] are characterized in Table 2.

### 4.3. Statistical Analysis and Data Visualization

The statistical analyses were conducted using STATISTICA v.13.5 software, TIBCO, Palo Alto, CA, USA. The normality of distribution was evaluated with the Shapiro–Wilk test. Continuous variables were presented as means and standard deviations. Categorical variables were presented as frequencies (percentages). Parametric data were compared using the independent *t*-test, and the categorical data were compared with Fisher’s exact test (two-sided). Statistical power analysis was performed for the statistically significant chi-square values. GraphPad Prism 10 software was used to create the graphs.

## 5. Conclusions

In conclusion, we found no association between substantia nigra echogenicity and mutations in the *DJ-1*, *SCNA*, *LRRK2*, *PRKN*, and *PINK1* genes that cause Parkinson’s Disease. In our cohort, PD patients with SN hyperechogenicity may exhibit RBD more frequently compared to those without SN hyperechogenicity, have a later onset of the disease, and are more often from rural areas. While transcranial ultrasonography is a valuable diagnostic tool for Parkinson’s Disease, it may not effectively identify patients at a higher risk of monogenic forms of the disease. However, these conclusions should be interpreted very cautiously. As we have stated above, the study has several limitations, and further analysis with more genetic variants should be conducted.

## Figures and Tables

**Figure 1 ijms-26-05492-f001:**
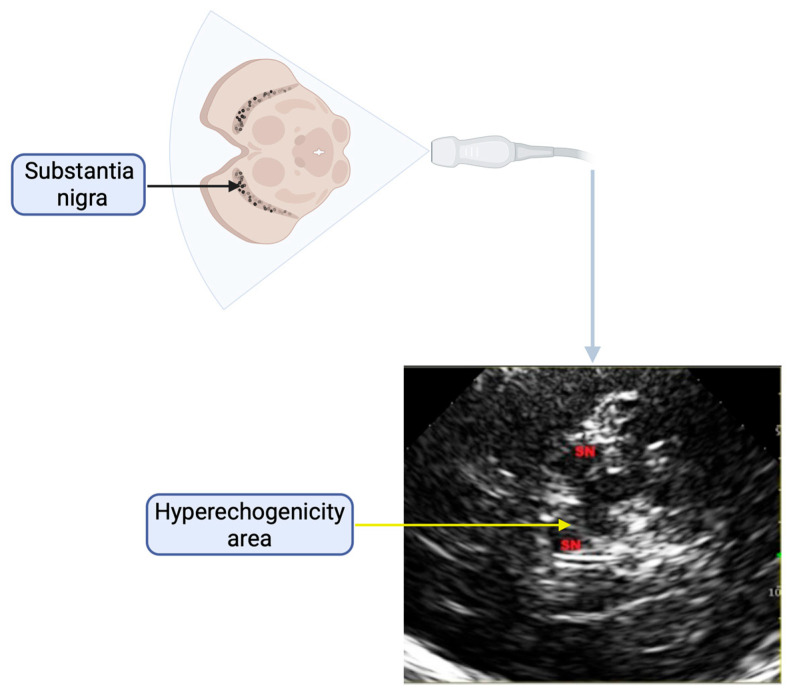
The figure presents a schematic representation of the transcranial sonography (TCS) of the substantia nigra. The ultrasound probe is positioned to visualize a transverse section of the brainstem, highlighting the anatomical location of the substantia nigra, where a region of increased echogenicity is commonly observed in patients with Parkinson’s Disease. Imaging artifacts, which may mimic the hyperechogenicity of the substantia nigra, should be considered during TCS image interpretation.

**Figure 2 ijms-26-05492-f002:**
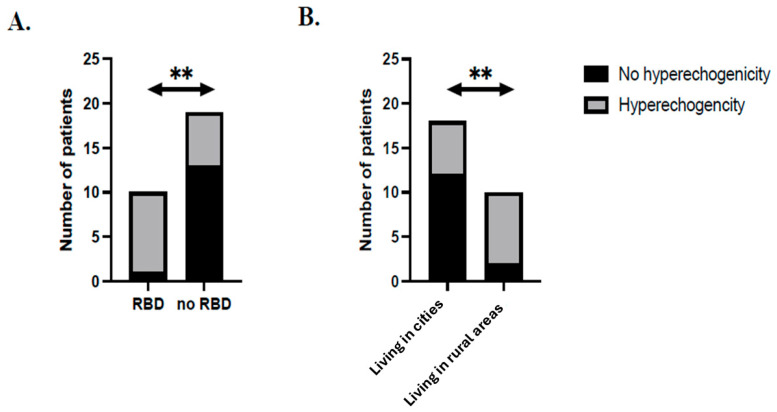
The association of substantia nigra hyperechogenicity with rural living and REM sleep behavior disorder (**A**): there was a significant difference in the number of patients living in rural areas with or without substantia nigra hyperechogenicity (** *p* < 0.05, Fisher’s exact test). (**B**): A significant difference was found in the number of patients with REM sleep behavior disorder with or without substantia nigra hyperechogenicity (** *p* < 0.05, Fisher’s exact test).

**Table 1 ijms-26-05492-t001:** Genetic, gender (total number of patients, N = 79), and other clinical data (total number, N = 29) in patients with and without hyperechogenic substantia nigra.

Variable	Hyperechogenicity (N = 48)	No Hyperechogenicity (N = 31)	*p* Value
Pathogenic mutations (*LRRK2*, *DJ1*, *PRKN*, *PINK1*, *SNCA*)	1 (2.1%)	2 (6.5%)	0.558
Gender (males)	33 (67.4%)	15 (50%)	0.157
Family history	11 (73.3%)	8 (57.1%)	0.450
Dyskinesia	5 (33.3%)	4 (28.6%)	1.000
Fluctuations	6 (40%)	4 (28.6%)	1.000
Tobacco smoking	6 (40%)	7 (50%)	0.819
Alcohol	3 (20%)	2 (14.3%)	1.000
Place of residence (city)	6 (42.9%)	12 (85.7%)	0.020
RLS	5 (33.3%)	4 (28.6%)	1.000
Depression	3 (20%)	5 (35.7%)	0.678
Dementia	1 (6.7%)	2 (14.3%)	1.000
Dysarthria	5 (33.3%)	7 (50%)	0.704
Dysphagia	4 (26.7%)	2 (14.3%)	0.420
RBD	9 (60%)	1 (7.1%)	0.004
First symptom:			
-Bradykinesia	8 (53.3%)	3 (21.4%)	0.677
-Rigidity	11 (73.3%)	4 (28.6%)	0.656
-Rest tremor	9 (60%)	6 (42.9%)	0.435
First treatment:			
-Levodopa	13 (86.7%)	11 (78.6%)	0.480
-Other	12 (80%)	12 (85.7%)	1.000
Agonist usage (now)	1 (6.7%)	4 (28.6%)	0.327
Age of onset (years)	51.5 ± 13.7	40.9 ± 13.1	0.040
Age at study (years)	58.2 ± 13.4	48.3 ± 11.4	0.040
Disease duration (years)	6.8 ± 3.8	7.5 ± 7.7	0.730

RBD—REM sleep behavior disorder; RLS—restless leg syndrome.

**Table 2 ijms-26-05492-t002:** Pathogenic variants in PD genes identified in patients’ cohort, and genotypes of patients.

Gene	Variant	cDNA HGVS ^#^	HGVS Protein	HGMD Path. Class.	ClinVar * Path. Class.	ACMG Class.
*PRKN*	Ex2 del	c.(?_40)_(109_?)del	p.?	DM	pathogenic	-
	Ex4 del	c.(?_440)_(534_?)del	p.?	DM	pathogenic	-
	p.Glu79Ter	c.235G>T	p.Glu79Ter	DM	-	LPath
*LRRK2*	p.Asn1437His	c.4309A>C	p.Asn1437His	DM	not provided	VUS/LPath
**Patients’ genotypes**						
*PRKN*	c.[(?-40)_(109_?)del];[(?-440_534_?)del]; p.[?];[?]					
*PRKN*	c.[(?-40)_(109_?)del];[235G>T]; p.[?];[(Glu79Ter)]					
*LRRK2*	c.[4309A>C];[=]; p.[(Asn1437His)];[=]					

^#^ description based on MLPA analysis; * in case of deletion compared to structural variant (≥50 bps).

## Data Availability

Data are available on request due to restrictions, e.g., privacy or ethical concerns. The data presented in this study are available on request from the corresponding author. The data are not publicly available due to patient’s privacy.

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
