# Peer review of "Clinical and Genetic Characteristics of Parkinson’s Disease Patients with Substantia Nigra Hyperechogenicity"

_ijms, 2025, doi:10.3390/ijms26125492_

Round 1

Reviewer 1 Report

Comments and Suggestions for Authors

This study represents an important but incremental advancement in diagnosis of PD.  Although the data appears unremarkable, the investigation of a non-invasive technique to add to the current list of possible diagnostic methods is worthwhile.  My only comment is whether a figure or image can be included to explain/illustrate what substantia nigra echogenicity looks like by transcranial ultrasonography.  This would increase the interest/readability of the article.  Also can you explain the genetic hieroglyphics used, not familiar with them.

Author Response

Comments 1: Although the data appears unremarkable, the investigation of a non-invasive technique to add to the current list of possible diagnostic methods is worthwhile.  My only comment is whether a figure or image can be included to explain/illustrate what substantia nigra echogenicity looks like by transcranial ultrasonography

Thank you for the valuable comment. We added an illustration presenting transcranial sonography and hyperechogenicity of substantia nigra (Figure 1).

Comments 2: Also can you explain the genetic hieroglyphics used, not familiar with them.

We have also added the the following fragment that makes the genetic description more clear:” One compound heterozygote of PRKN was revealed in each group, in one patient with hyperechogenicity PRKNbiallelic, heterozygous mutations – Exon4 deletion and nonsense p.Glu79Ter variant and in one patient without hyperechogenicity heterozygous, biallelic Exon2 and Exon 4 deletions. The LRRK2 heterozygous missense variant p.Asn1437His was identified in one patient without hyperechogenicity (genotypes according to HGVS recommendation are described in Materials and Methods).”

Reviewer 2 Report

Comments and Suggestions for Authors

The paper titled” Clinical and genetic characteristics of Parkinson’s Disease patients with substantia nigra hyperechogenicity” provides a valuable clinical characterization of PD patients with SN hyperechogenicity, highlighting novel phenotypic correlations. However, its genetic analysis is constrained by small sample size and limited gene coverage, while its retrospective design and incomplete data affect the strength of associations. The proportion of sporadic PD patients is higher. The authors should explain why they did not analyze sporadic patients but instead chose those with genetic mutations. Additionally, it needs to be clarified whether this phenomenon of hyperechogenicity in the substantia nigra region applies to general sporadic patients. Despite these limitations, the study serves as a foundation for larger, prospective investigations into the interplay between SN hyperechogenicity, genetics, and environmental risk factors in PD.

Author Response

Comments 1: The paper titled” Clinical and genetic characteristics of Parkinson’s Disease patients with substantia nigra hyperechogenicity” provides a valuable clinical characterization of PD patients with SN hyperechogenicity, highlighting novel phenotypic correlations. However, its genetic analysis is constrained by small sample size and limited gene coverage, while its retrospective design and incomplete data affect the strength of associations. The proportion of sporadic PD patients is higher. The authors should explain why they did not analyze sporadic patients but instead chose those with genetic mutations. Additionally, it needs to be clarified whether this phenomenon of hyperechogenicity in the substantia nigra region applies to general sporadic patients. Despite these limitations, the study serves as a foundation for larger, prospective investigations into the interplay between SN hyperechogenicity, genetics, and environmental risk factors in PD.

Thank you for your valuable comment. The aim of the study was to compare PD patients with SN hyperechogenicity and those without. The presence of genetic variant was just one of the variable, which in this study does not reach statistical significance.

Reviewer 3 Report

Comments and Suggestions for Authors

Title: Clinical and genetic characteristics of Parkinson’s Disease patients with substantia nigra hyperechogenicity.

Summary

In this article, Lukasz Milanowski et al. investigated whether monogenic forms of Parkinson’s disease (PD) are more prevalent in PD patients with substantia nigra (SN) hyperechogenicity. Eighty-eight PD patients were included in this study and SN hyperechogenicity was identified in 48 patients. The results showed that the PD patients with SN hyperechogenicity did not have monogenic forms of PD more frequently compared to the non-hyperechogenicity group. This study indicated no association between substantia nigra echogenicity and mutations in the SNCA, 21 LRRK2, DJ1, PRKN, and PINK1 genes.

This manuscript is not suitable for publication in the International Journal of Molecular Sciences.

Major comments are given below.

  1. The sample size is too small to make valid conclusions. More patients should be included in this study.
  2. The authors should specify what mutations of SNCA, LRRK2, DJ1, PRKN, and PINK1 genes are studied in this article? Why these mutations are important for this study?
  3. The data presented in the current manuscript is too weak to support its conclusions.

Author Response

Comments 1: The sample size is too small to make valid conclusions. More patients should be included in this study.

We agree, that sample size is small but the statistical power of revealed clinical observations were sufficient. We introduced to the methodology section results section sentence:” Statistical power analysis was performed for the chi-square statistically significant values.” and in results section sentence “The calculated power for the statistically significant analysis for the chi-square tests were at least 99%”

Comments 2: The authors should specify what mutations of SNCA, LRRK2, DJ1, PRKN, and PINK1 genes are studied in this article? Why these mutations are important for this study?

We performed the statistical analysis for the following genes as stated in the methodology section: full-length coding sequences and dose analysis of the genes PRKN (Exons 1–12), PINK1 (Exons 1–8), DJ-1 (Exons 1–6), SNCA (Exons 1-6), and the diagnostically recommended exons of the LRRK2 gene (Exons 30, 31, 34, 35, 41, 48) were conducted for all patients. We have added in the methodology section:” The chosen genes and variants are most commonly identified in the PD patients, especially in Polish population”

Comments 3: The data presented in the current manuscript is too weak to support its conclusions.

Thank you for these comments. Based on the statistical power analysis the results support the conclusion. However, we rewrite them that they are now more cautious: “However, the conclusions should be interpreted very cautious. As we stated above, the study has several limitations and further analysis with more genetic variants should be introduced”

Round 2

Reviewer 3 Report

Comments and Suggestions for Authors

This manuscript is not publishable in IJMS. More data should be provided and presented in the manuscript.

Author Response

Thank you for the valuable comments. We have expanded the article and now should meet the publication rigors